# Real-Time Polymerase Chain Reaction: Current Techniques, Applications, and Role in COVID-19 Diagnosis

**DOI:** 10.3390/genes13122387

**Published:** 2022-12-16

**Authors:** I Made Artika, Yora Permata Dewi, Ita Margaretha Nainggolan, Josephine Elizabeth Siregar, Ungke Antonjaya

**Affiliations:** 1Department of Biochemistry, Faculty of Mathematics and Natural Sciences, Bogor Agricultural University, Bogor 16680, Indonesia; 2Eijkman Research Center for Molecular Biology, National Research and Innovation Agency, Bogor 16911, Indonesia; 3Emerging Virus Research Unit, Eijkman Institute for Molecular Biology, Jalan Diponegoro 69, Jakarta 10430, Indonesia; 4Eijkman Oxford Clinical Research Unit, Eijkman Institute for Molecular Biology, Jalan Diponegoro 69, Jakarta 10430, Indonesia

**Keywords:** polymerase chain reaction, real-time PCR, quantitative PCR, molecular diagnosis, COVID-19, SARS-CoV-2

## Abstract

Successful detection of the first SARS-CoV-2 cases using the real-time polymerase chain reaction (real-time PCR) method reflects the power and usefulness of this technique. Real-time PCR is a variation of the PCR assay to allow monitoring of the PCR progress in actual time. PCR itself is a molecular process used to enzymatically synthesize copies in multiple amounts of a selected DNA region for various purposes. Real-time PCR is currently one of the most powerful molecular approaches and is widely used in biological sciences and medicine because it is quantitative, accurate, sensitive, and rapid. Current applications of real-time PCR include gene expression analysis, mutation detection, detection and quantification of pathogens, detection of genetically modified organisms, detection of allergens, monitoring of microbial degradation, species identification, and determination of parasite fitness. The technique has been used as a gold standard for COVID-19 diagnosis. Modifications of the standard real-time PCR methods have also been developed for particular applications. This review aims to provide an overview of the current applications of the real-time PCR technique, including its role in detecting emerging viruses such as SARS-CoV-2.

## 1. Introduction

The polymerase chain reaction (PCR) was first used to amplify particular DNA sequences and has since been extended into one of the most robust research tools in biological sciences and medicine. Its extension to RNA studies was based on using a reverse transcriptase enzyme to first make complementary DNA (cDNA) and then employing this in the process of PCR amplification, a method termed reverse transcription PCR (RT-PCR) [1]. However, as the standard PCR cannot be reliably used for accurate quantification, the technique was refined, giving the powerful analytical tool we now call real-time polymerase chain reaction (real-time PCR) [2].

At the end of 2019, the COVID-19 pandemic, due to the novel SARS-CoV-2, hit the globe and gave rise to a great challenge to public health laboratories. The gold standard diagnosis for SARS-CoV-2 infection is a nucleic acid amplification test (NAAT), and real-time PCR assay is the major platform that was applied [3]. COVID-19 also forced Indonesia to increase the number of laboratories with the capacity for COVID-19 detection. In the beginning, the government assigned only one lab. However, due to the increasing number of COVID-19 cases, by 29 April 2020, as many as 89 laboratories were officially appointed [4]. The fact that the real-time PCR platform is a multipurpose platform and can be applied in various fields of application is worthy of exploration. The technique can be used for basic molecular research right through to an approved molecular diagnostic assay. The exploration of the current wide range of applications of the real-time PCR method is critical, including its feasibility in low-middle income countries.

## 2. Basic Principles

Real-time polymerase chain reaction (real-time PCR), also known as quantitative PCR, is a modification of the PCR strategy which allows monitoring of the PCR progress in real-time PCR itself is an enzymatic process used in vitro for the amplification of a selected DNA region through several orders of magnitude, generating thousands to millions of copies of a specific DNA segment. Ingredients needed include template DNA, primers, nucleotides (dNTPs), and thermostable DNA polymerase [5,6]. In addition to improved accuracy, sensitivity, and rapidity, one of the principal advantages of the real-time PCR over basic PCR is that this technique provides a reliable quantification relationship between the number of starting target sequences (before the amplification by PCR) and the amount of amplicon accumulated in a particular PCR cycle [5]. This is of paramount importance for the precise quantification of the target nucleic acids, which is critical for mRNA quantification in gene expression analysis [7] and the determination of the viral load of a clinical specimen [8]. Moreover, there is no need for post-PCR processes, thus minimizing the chance of cross-contamination due to previous amplicons [5]. This real-time PCR technique, therefore, has revolutionized the detection and quantification of target nucleic acids and gained a wide range of applications [9].

### 2.1. Quantification

The number of DNA molecules available in the starting mixture determines the quantity of amplicon generated following a set number of PCR cycles. If only a few DNA molecules are present at the start of the PCR process, relatively little amplicon will be synthesized. On the contrary, if there are large amounts of starting molecules, then the amount of product will be higher. This relationship permits the use of PCR to calculate the number of DNA molecules present in samples by measuring the amount of product that is generated. However, using conventional PCR, in which the amplicons are measured after finalizing the PCR process (end-point detection), the quantitative correlation between the starting DNA molecules and the PCR product becomes imprecise as large differences in the number of starting DNA cause relatively small differences in the resulting PCR products. This is due to factors such as the presence of inhibitors of the polymerase reaction, reagent limitation, and the accumulation of pyrophosphate molecules. The ability to monitor the PCR product in real-time, especially during the exponential phase, makes real-time PCR a reliable quantitative method because, during this phase of the PCR reaction, a precise quantitative relationship between the amount of starting DNA and the quantity of PCR product can be established. By detecting the amount of amplicon during the exponential phase, it is possible to extrapolate back to the quantity of the starting DNA in the mixture, hence, the concentration of the nucleic acids in the original sample [2,5].

Plotting the amount of PCR product (amplicon) versus the number of reaction cycles produces a representative real-time PCR amplification curve, as illustrated in Figure 1. Major phases of the amplification curve include linear (at the start), exponential (logarithmic-linear), and plateau phases. Throughout the initial cycles of the PCR process, the values of the fluorescence emission of the product represent the linear ground phase and do not exceed the baseline. During the exponential phase, PCR gains its optimum amplification period, doubling the product after each cycle. The ideal reaction conditions are achieved during this phase, with none of the reaction components being limiting. Fluorescence intensity in the exponential phase is used for data calculation. Although theoretically, PCR itself is an exponential process, and the number of DNA molecules should double after each cycle because reaction components eventually become limiting, so the rate of target amplification decreases, and the PCR reaction reaches a plateau. The fluorescence intensity at the plateau phase is, therefore, not useful for data calculation [5,10].

As shown in Figure 1, there are several terms used related to the amplification curve of real-time PCR. The baseline is defined as the number of PCR cycles in which a fluorescent reporter signal accumulates but is below the limits of detection. Threshold refers to an arbitrary value selected based on the variability of the baseline to reflect a statistically significant increase of signal over the baseline, hence distinguishing a relevant amplification signal from the background. It is generally set at 10× the standard deviation for the average signal of the baseline fluorescence. A fluorescent signal detectable above the threshold is assumed to be a real signal used to define the threshold cycle (Ct) for a sample. Ct refers to the fractional PCR cycle number in which the reporter fluorescence level is higher than the minimum detection level, the threshold. The availability of more nucleic acid templates at the beginning of the reaction results in fewer cycles required to reach the position at which the fluorescent signal is substantially higher than the background. Nucleic acid quantification can then be performed by comparing the Ct values of the samples at a particular fluorescence value with similar data obtained from a series of standards by constructing a standard curve [5,11,12]. A standard curve can be generated based on a serial dilution of a starting amount of known nucleic acids, such as a plasmid for the gene of interest or a chemically synthesized single-stranded sense oligonucleotide for the whole amplicon. Alternatively, a standard curve can also be generated based on a cell line with a known copy number or expression level of the gene of interest. In the absence of standard curves, relative quantification can be carried out by comparing the Ct values of the samples with that of a reference control [5].

Theoretically, real-time PCR can only be applied to the amplification of templates in the form of DNA molecules. How, then, to detect and quantify an RNA sample? For these purposes, the RNA molecule is first reverse-transcribed into a complementary DNA (cDNA) using reverse transcriptase, followed by conversion of the generated single-stranded cDNA to double-stranded DNA. This double-stranded DNA is then amplified using standard PCR. This procedure is known as real-time reverse transcription polymerase chain reaction (real-time RT-PCR) [6]. The real-time RT-PCR can be carried out using either a one-step or a two-step method. In one-step real-time RT-PCR, the RT step is coupled with PCR. In this process, RNA is reverse transcribed to cDNA and then amplified in one reaction. The main advantages of this method are rapidity of set-up, cheapness, and involving less handling of samples to reduce pipetting errors and contamination. However, as this method employs gene-specific primers for both the RT and PCR occurring in one reaction tube, other genes of interest cannot be amplified for later analysis [13]. In two-step real-time RT-PCR, the process consists of two separate steps. The initial step is an RT reaction to construct cDNA. The second step is the cDNA amplification using traditional real-time PCR. The main advantage to two-step RT-PCR is that the cDNA is typically generated using random hexamer- or oligo-dT primers, which allow complete conversion of the messages in the RNA sample into cDNA, hence, permitting future analysis of other genes [13].

### 2.2. Probes

Real-time PCR systems employ a fluorescent reporter of the probe for detection and quantification. In general, they are classified into two main groups depending on the fluorescent agent used and the specificity of the PCR detection. The first class is based on double-stranded DNA intercalating molecules such as SYBR Green I and EvaGreen, allowing the detection of both specific and non-specific amplicons. For the second group, fluorophores are linked to oligonucleotides. Thus, they only detect specific amplicons [14]. This group includes hydrolysis probes (such as the TaqMan probe), dual hybridization probes, molecular beacons, and scorpion probes [5]. Other types of probes, such as those which belong to analogs of nucleic acids, have also been described [14]. A fluorophore is a fluorescent molecule that absorbs light energy at a particular wavelength and then re-emits light at a longer wavelength. There are two kinds of fluorophores: donor or reporter and acceptor or quencher. If a donor fluorophore absorbs light energy, it raises its energy level to that of an excited state. The process of a return to the ground state is accompanied by the emission of energy as fluorescence. This emitted light energy can be transmitted to an adjacent acceptor fluorophore when the two fluorophores are present in proximity. This transfer of excited-state energy from a fluorescence-reporter to a quencher is termed “fluorescence-resonance-energy transfer” (FRET) [14]. It should be noted that there are two distinct FRET mechanisms depending on how the energy is passed on to the acceptor fluorophore and dissipated, called FRET-quenching and FRET. The phenomenon of FRET quenching occurs when the energy of the quencher (a non-fluorescent molecule) is released as heat rather than emitted as light. FRET happens when the transferred energy is emitted as fluorescent light due to the acceptor molecule being a fluorocompound [14].

SYBR Green 1 is the most commonly used double-stranded DNA intercalating agent. It is a dye that attaches to the minor groove of double-stranded DNA, regardless of its sequence. It only fluoresces when inserted into double-stranded DNA, as illustrated in Figure 2. The strength of the fluorescence signal is therefore dictated by the quantity of double-stranded DNA existing in the reaction. The superiorities of SYBR Green 1 are low cost, convenience, and sensitivity. The major drawbacks of this probe are that they are not specific because the probe interacts with all double-stranded DNAs synthesized in the course of the PCR process, including the nonspecific amplicons and primer-dimers [5,14]. Considering that nonspecific products, including primer-dimers, are able to be generated during the PCR process, it is highly recommended to perform a melting curve analysis to determine the specificity of the amplified DNA sequences [14]. Notably, by optimizing the SYBR Green technique, its performance and quality can be as good as the specific TaqMan assay [15]. Other DNA-binding dyes available commercially include ethidium bromide, YO-PRO-1, SYBR® Gold, SYTO, BEBO, BOXTO, and EvaGreen [14]. The SYBR Green has recently been employed as a probe in a quantitative PCR platform to detect SARS-CoV-2 [16].

The TaqMan Probe is a very popular hydrolysis probe, which is designed to attach to a specific sequence of the target DNA. The mechanism of its action depends on the 5′–3′ exonuclease activity of Taq polymerase, which hydrolyzes the attached probe throughout PCR amplification. The TaqMan probe has a fluorescent reporter dye linked to its 5′ end and a quencher dye at its 3′ terminus. While the probe is intact, the reporter and quencher stay in close proximity, and excitation energy are quenched, prohibiting the emission of any fluorescence. In the presence of the target sequence, the TaqMan probe binds downstream from one of the primer sites. During PCR, when the polymerase replicates a DNA sequence on which a TaqMan probe is bound, the 5′ exonuclease activity of the polymerase cuts the probe. This sets apart the fluorescent and quenching dyes, and excitation energy is released as fluorescent light, as illustrated in Figure 3. Fluorescence intensity increases in each cycle in proportion to the rate of cleavage of the probe [5,14]. The TaqMan probe has been used to develop a multiplex real-time PCR method for the concurrent detection of novel swine coronaviruses to improve animal and public health [17].

The dual hybridization probe system consists of two hybridization probes. One carries a donor fluorophore at its 3′ terminus, and the other harbors an acceptor fluorophore at its 5′ end. Following the denaturation step, both probes hybridize to their target sequence in a head-to-tail formation during the annealing step. This makes the two dyes in close proximity mediating the energy transfer process (FRET). The donor dye in one of the probes absorbs light. It transmits energy, permitting the other one to dissipate that energy as fluorescence at a higher wavelength, as illustrated in Figure 4. As the fluorescence from the acceptor probe only happens if both the donor probe and the acceptor probe anneal to the PCR product, the detected fluorescence is directly proportional to the amount of DNA formed during the PCR process. The specificity of this reaction is therefore increased because a fluorescent signal is only happened upon two independent probes hybridizing to their specific target sequence [5,18]. The dual hybridization probe has been applied in a real-time PCR technique for rapid identification of *Bacillus anthracis* in environmental swabs based on the amplification of a special chromosomal marker, the E4 sequence. The method may contribute to strengthening the biodefense system [19].

The molecular beacon is another hybridization-based probe suitable for real-time PCR. This probe also contains attached fluorescent and quenching dyes at either end of a single-stranded DNA molecule. However, it is intended to form a stem-and-loop structure when free in solution so as to bring the fluorescent dye and the quencher in close proximity, and, as a result, resonance energy is quenched. The loop segment of the molecule is complementary to the target nucleic acid molecule, and the stem is formed by the annealing of complementary arm sequences on the termini of the probe sequence. When the probe sequence in the loop attaches to a complementary nucleic acid target sequence during the annealing step, a conformational change takes place that forces the stem apart. This leads to the separation of the fluorophore from the quencher dye. Hence, as illustrated in Figure 5 [5], resonance energy is emitted as light. Unlike the TaqMan probe, the molecular beacon probe does not require a polymerase with exonuclease activity [20]. The molecular beacon probe has recently been used in a real-time PCR assay for the detection of SARS-CoV-2 [21].

The scorpion probe is another fluorescence-based method developed for the specific detection of PCR products. Similar to molecular beacons, the scorpion probe adopts a stem-and-loop configuration due to the presence of complementary stem sequences on the 5′ and the 3′ sides of the probe. A fluorophore reporter molecule is attached to the 5′ end and a quencher molecule is joined to the 3′ end of the probe (Figure 6). The specific probe sequence is kept within the hairpin loop, linked to the 5′ terminus of a PCR primer sequence by a non-amplifiable monomer called a PCR stopper. The function of the PCR stopper is to prevent PCR from amplifying the stem-loop sequence of the scorpion primer. During PCR, scorpion primers are extended to generate amplicons. During the annealing phase, the specific probe sequence in the scorpion tail curls back to hybridize with the complementary target sequence in the amplicon. This hybridization event opens up the hairpin loop and prevents the reporter molecule’s fluorescence from being quenched, and therefore a light signal is emitted. As the tail of the scorpion and the amplicon become part of the same strand of DNA, the interaction is intramolecular. This is beneficial as it leads to an effective instantaneous reaction giving a much stronger signal compared with the bimolecular interaction used in TaqMan or molecular beacon techniques [5]. The scorpion probe has been employed in a real-time PCR method to detect *Escherichia coli* in dairy products for food safety monitoring [22].

## 3. Applications

Apart from offering great sensitivity and specificity, real-time PCR can be applied for both qualitative and quantitative analysis. Therefore, it has become the method of choice for the rapid and sensitive detection and quantification of nucleic acid in biological samples for many diverse applications such as gene expression analysis, detection of mutation, determination of cancer status, microRNA analysis, detection of genetically modified organisms, bacterial detection, bacterial quantification, viral detection, and viral load measurement. Due to its versatility and usefulness, the real-time PCR technique has been employed in many research areas, including biomedicine, microbiology, veterinary science, agriculture, pharmacology, biotechnology, and toxicology [14]. Selected examples of the application of real-time PCR are presented in Table 1.

### 3.1. Analysis of Gene Expression

Reverse-transcription quantitative PCR (RT-qPCR) has become a popular technique to quantify gene expression because it is efficient, simple, and low-cost. It is a general test to determine the amount of expression of target genes in a wide range of samples from different sources, such as in tissues, blood, and cultured cells originating from bacteria, plants, animals, and humans. It is important to note that for reliable transcriptional quantification, the relative expression of a particular target gene is calculated based on the use of reference gene(s) as endogenous control(s), which exhibit a constant expression throughout the experimental conditions. The inclusion of endogenous reference (housekeeping) genes in the assay serves as an internal reaction control to normalize mRNA levels between different samples in order to allow for an exact comparison of the level of mRNA transcription [49,50,51].

It is critical to select a suitable reference gene for each experiment. An ideal reference gene for RT-qPCR should not be affected by the experimental conditions and the level of expression [49]. For gene expression analysis in a human cell line, it was found that the polyubiquitin-C gene (*UBC*) and DNA topoisomerase 1 gene (*TOP1*) show the least variation and the highest expression stability among the twelve most commonly used human reference genes [49]. In other studies, the expression of the cyclophilin A gene (PPIA) was found to be most stable in human airway epithelial cells [52]. Some of the commonly used reference genes in the study of gene expression are presented in Table 2.

The real-time RT-PCR technique was implemented to investigate the non-thermal effects of wireless fidelity (Wi-Fi) radiofrequency radiation on the expression of selected genes of bacteria to confirm a global gene expression study carried out by using high-throughput RNA-sequencing. The target genes included *pgaD*, *fiC*, *cheY*, *malP*, *malZ*, *motB*, *alsC*, *alsK*, *appB,* and *appX,* together with housekeeping genes *gyrA* and *frr* employed for gene normalization [23]. Total RNA was extracted from bacterial cells and followed by the synthesis of cDNA. A real-time PCR test using specific primers for every reaction was then performed. It was found that the results from real-time RT-PCR assays were consistent with that obtained from RNA sequencing [23]. The real-time RT-PCR method has also been applied to analyze gene expression of the plant *Arabidopsis thaliana* ATP-binding cassette (ABC) transporters to screen candidates of a monolignol-transporter which transports monolignols from the cytoplasm to the cell wall for lignin biosynthesis [24]. Total RNA was isolated from several plant organs, followed by cDNA synthesis from each RNA sample using a mixture of oligo (dT) and random primers. Each cDNA generated was used as a template for real-time PCR analysis. The expression of target transporter genes (*ABCG29*, *ABCG30*, *ABCG33*, *ABCG34*, and *ABCG37*) of wild-type and mutant plants were analyzed in comparison to reference genes. The RT-qPCR technique was able to resolve the expression level of each target gene. It was concluded that each member of the multiple gene systems is expressed in the process of lignin synthesis [24].

The real-time RT-PCR technique was recently applied to measure expression levels of *CPEB4*, *APC*, *TRIP13*, *EIF2S3*, *EIF4A1*, *IFNg*, *PIK3CA,* and *CTNNB1* genes in tumors and peripheral blood samples of colorectal cancer patients in stages I–IV of the disease [25]. Total RNA was extracted from tissues or peripheral blood samples, followed by reverse transcription to produce cDNA. Using specific primers for each gene, real-time PCR was then performed to analyze the mRNA level of each gene in colorectal cancer tissue specimens, colorectal cancer blood samples, normal colon tissues, and normal blood samples. The study concluded that TRIP13 and CPEB4 mRNA up-regulation in the peripheral blood of patients with colorectal cancer might be a potential target for an early-stage test of colorectal cancer [25]. Similarly, the real-time RT-PCR method was employed to determine and evaluate the microRNAs (miR-150, miR-146a, hsa-let-7e) expression profile within peripheral blood mononuclear cells (PBMCs) infected with the dengue virus. Total RNA was isolated from dengue virus-infected PBMCs, followed by real-time RT-PCR assay. Data showed that dengue viral infection upregulates microRNA expression. Notably, microRNAs play roles in regulating the expression of cytokine genes in response to dengue viral infection [26].

### 3.2. Detection of Mutation

In addition to its wide application in gene expression analysis, real-time PCR is regarded as a simple, robust, and highly selective method for detecting mutation [56]. A widely employed approach to detect DNA sequence variants is the use of one or both oligonucleotides designed to attach at the sites of sequence variation. The use of a primer whose sequence matches a particular variant is intended to selectively amplify only the variant, although, in practice, mismatched amplification may occur. The amount of this non-specific amplification varies widely depending on the particular base mismatch between the allele-specific primer and the wild-type sequence [56]. A simple and robust real-time PCR method has been applied to detect *PIK3CA* mutations, the most common driver mutations in human breast cancer [27]. The assay employed a set of primers specifically designed to target the mutant sequence while minimizing the synthesis of mismatched products derived from the wild-type allele. Antisense oligonucleotide targeting the mutant-specific sequence with a variant base located at its 3′ end was used to reduce cross-amplification of the wild-type template. Moreover, a non-productive phosphate-modified oligonucleotide complementary to the wild-type sequence was employed to suppress the amplification of the wild-type allele [27]. Similarly, a highly sensitive and specific RT-qPCR method has been developed for screening BRAF V600E/K mutation, which frequently occurs in lung cancers. The technique is useful for studying the incidence and clinicopathological features of BRAF V600E/K mutation in lung cancer patients [28].

The real-time PCR technique has also recently been applied to quantitatively detect hepatitis B virus (HBV) M204V mutation [29]. This is an amino acid substitution in the hepatitis B viral polymerase linked to viral resistance to nucleotide analogs, the main treatment option for patients suffering from chronic hepatitis B. For quantitative measurement, a plasmid carrying the M204V mutation was synthesized. The method showed advantages in terms of sensitivity, specificity, and efficiency in detecting HBV M204V mutations and provided a new option for monitoring drug resistance [29]. A mismatch amplification mutation assay for rapid detection of *Neisseria gonorrhoeae*, the causative agent of the sexually transmitted infection gonorrhea, has been developed using a real-time PCR platform. The assay was also designed to rapidly detect antimicrobial resistance determinants in clinical samples. The strategy was considered promising to detect *N. gonorrhoeae* and infer antimicrobial resistance directly in genital specimens [30].

### 3.3. Food Analysis

Effective detection of a genetically modified organism (GMO) is critical for regulatory enforcement, traceability in terms of biosafety, environmental impact, socio-economic consequences, and for protecting consumer freedom of choice [57,58]. Real-time PCR is the most common strategy for GMO detection, identification, and quantification. The technique is applicable for both unprocessed and processed food/feed matrices. The most common transgenic elements targeted include p35S (35S promoter from cauliflower mosaic virus), tNOS (nopaline synthase terminator from *Agrobacterium tumefaciens*), and some markers such as Cry3Bb, gat-tpinII, t35S pCAMBIA, and taxon-specific markers [57]. By targeting the p35S and tNOS, a highly sensitive real-time PCR-based GMO detection was developed using a large number of DNA templates capable of detecting a great variety of different GMOs, including some uncertified ones. The method was claimed to be the most sensitive method for the detection of genetically engineered maize. Importantly, the technique was able to detect genetically modified maize in the form of both raw grain and processed foods [31]. Recently, a systematic real-time PCR array combined with a prediction system for rapid tracking of genetically modified soybeans has been developed. A total of 16 promoters, 15 terminators, and 21 genes were employed for the development of the screening assays [32]. The genetic elements targeted include p35S, tNOS, pRbcS4, tE9, pat gene, and lectin gene. The method has been successfully tested using 17 genetically modified soybean events and 23 processed foods and could be applied to trace the absence or presence of genetically modified soybean events [32]. Real-time PCR can also be utilized to detect unauthorized genetically engineered microorganisms by targeting the *cat*, *aadD* or *tet-l* genes [33].

Recently, a real-time PCR-based method for testing allergens in food was developed by targeting three chloroplast markers (mat k, rpl16, and trnH-psbA) and a nuclear low-copy target (the Ara h 6 peanut allergen-coding region) [34]. It was found that the mat k marker gave the most sensitive and efficient detection for peanuts [34]. Furthermore, the technique has been employed for the detection of pork in meat-based food products by using specific primers targeting the mitochondrial cytochrome-b gene. Notably, pork is considered non-halal (prohibited from eating according to Islamic law) for Muslim communities, and therefore accurate labeling of meat-based products is essential [35].

### 3.4. Bioremediation Monitoring

The real-time PCR technique has been applied as a cultivation-independent method to monitor microbial biodegradation of contaminants and pollutants by determining the occurrence and abundance of microbial-specific gene markers, which reflect the biodegradation potential and efficiency. The real-time PCR method was implemented to monitor the dynamics of the crude oil-degrading bacterium *Nocardia* sp. H17-1 in the course of bioremediation of crude-oil-contaminated soil by detecting and quantifying the genes 16S rRNA (encoding 16S ribosomal RNA), *alkB4* (specifying alkane monooxygenase), and 23CAT (encoding catechol 2,3-dioxygenase) [36]. Microbial-based degradation of contaminants and pollutants is a process having economic and environmental benefits, and the monitoring of the operation is critical to ensure that the introduced microorganisms are effective and can survive in harsh conditions. The real-time PCR technique is preferred when compared to the cultivation-dependent methods, such as the plate count method, as most (more than 99%) of the microbes in the environment cannot be cultivated. In addition, the culture-based method is laborious and lacks the specificity and sensitivity required to track the inoculants accurately [36,37]. Real-time PCR can also be applied for rapid detection of aniline-degrading bacteria such as *Acidovorax* sp., *Gordonia* sp., *Rhodococcus* sp., and *Pseudomonas putida* in activated sludge. Of note, aniline and its derivatives are important environmental pollutants due to their significant toxic and mutagenic effects [38]. In addition, the technique has been applied to develop methods for the quantification of *Methanoculleus*, *Methanosarcina,* and *Methanobacterium* in anaerobic digestion, a growing platform for bioenergy production from wet biomass waste [39].

Recently, a novel method termed digital PCR (dPCR) has been developed and is considered superior compared to traditional real-time PCR in terms of accuracy, sensitivity, precision, and reproducibility for microbial biodegradation monitoring [37]. The technique is suitable for detecting low-copy targets, environmental DNA, rare alleles, minor mutations, and the analysis of methylated DNA. The dPCR approach enables absolute quantification of target nucleic acids without the requirement for standard curves. The technique relies on a partition of the assembled reaction into enormous independent PCR sub-reactions. PCR amplification is carried out to its endpoint, and absolute quantification of target molecules is performed following Poisson distribution, which allows accurate quantification of target molecules [37]. Alternatively, microbial dynamics during contaminant biodegradation can also be analyzed using shotgun metagenomics and metatranscriptomics approaches. Cao and coworkers applied metagenomics and metatranscriptomics analysis as an emerging tool to study the whole picture of microbial functions and activities in the biodegradation of naturally and chemically dispersed marine diluted bitumen using artificial, experimental ecosystems termed “microcosms” to simulate the natural marine environment in the laboratory [59]. It was concluded that the metagenomics and metatranscriptomics strategies could be used to obtain a broad overview of microbial metabolic functions and activities for diluted bitumen degradation [59]. Based on 16S rRNA gene amplification and sequencing data, a better representation of the marine environment microbial communities was achieved using a larger scale of microcosms due to increased biomass available for deep sequencing [59]. Another powerful emerging method, called microfluidic technology, has also been developed, which enables biological and biochemical assays of microbes to be performed in very small volumes within a well-defined microenvironment mimicking their natural habitats [60].

### 3.5. Detection and Quantification of Pathogen

A multiplex real-time PCR assay has also been designed and validated for simultaneous detection at a high level of specificity for several bacterial pathogens causing pneumonia [40]. The target bacteria include *Klebsiella pneumoniae*, *Pseudomonas aeruginosa*, *Staphylococcus aureus*, and *Moraxella catarrhalis*. The sequence of primers was intended to bind a specific gene in each pathogen, which included *yphG* (encoding an uncharacterized protein, YphG) for *K*. *pneumonia*, *regA* (specifying exotoxin A regulatory protein) for *P. aeruginosa*, *nuc* (encoding micrococcal nuclease) for *S. aureus* and *copB* (specifying outer membrane protein B2) for *M. catarrhalis* [40]. The multiplex real-time PCR assay could also be applied for rapid identification and quantitative analysis of microbial species, such as *Aspergillus* species [41]. Primers were designed to target the *BenA* (encoding protein BenA) and *cyp51A* (encoding cytochrome P450 14-alpha sterol demethylase) genes. The assay was reported to show 100% specificity to every *Aspergillus* section (*Fumigati*, *Nigri*, *Flavi*, and *Terrei*) without cross-reaction between different sections. In quantitative analysis, the assay showed a limit of detection (LOD) and limit of quantitation (LOQ) of 40 fg and 400 fg, respectively [41]. In addition, a real-time RT-PCR technique was employed as a tuberculosis molecular bacterial load assay (TB-MBLA) to quantify *Mycobacterium tuberculosis* bacillary loads using primers targeting the bacterial 16S rRNA [42]. This RNA molecule was a preferred target for detection because DNA is a stable molecule that survives long after cells have died and hence is not a good standard for calculating life cells which are crucially critical for evaluating a treatment response [42]. The real-time PCR method was also employed to determine the growth fitness of plasmodium mutants that are resistant to atovaquone by analyzing the level of the parasite mitochondrial DNA [43].

Pathogenic viruses such as Chikungunya virus (CHIKV) [45], Zika virus (ZIKV) [61], human adenoviruses [46], and others have been identified using the real-time PCR approach. For CHIKV detection, viral RNA was isolated and used as a template for CHIKV quantitative RT-PCR [45] using primers targeting the nonstructural protein 1 gene [44]. Similarly, ZIKV-specific real-time RT-PCR can also be applied to provide evidence of ZIKV infection [61] using primer sets specific to particular sequences within the ZIKV genome [62]. A practical in-house real-time PCR assay was developed for the detection of human adenovirus from viral swabs [46]. In this assay, the viral DNA was extracted from specimens using a combination of homogenization and heat treatment. The real-time PCR was carried out as duplex reactions using primers and probes designed to target and detect the adenovirus hexon gene and an exogenous internal control (pGFP) [46]. In addition, the real-time PCR assay has been used to analyze viral load to study the viremic profile in chikungunya-infected patients [8]. Similarly, the technique was applied to determine viral load during the acute phase of chikungunya infection in children. Viral RNA was extracted from plasma samples and used as a template for quantitative RT-PCR targeting a 200 bp region of the envelope (E1) gene [47]. Recently, the real-time RT-PCR technique was employed to detect and quantify SARS-CoV-2 in specimens collected from COVID-19 suspects or persons in contact tracing programs [48,63,64].

## 4. Detection and Quantification of SARS-CoV-2

Together with the DNA sequencing method, the real-time RT-PCR technique was successfully applied to detect and identify for the first time the newly emerged 2019 novel coronavirus (2019-nCoV) in Wuhan, China in December 2019 by employing primers that targeted a consensus RNA-dependent RNA polymerase (RdRp) region of pan β-CoV [65]. The *Coronaviridae* Study Group of the International Committee on Taxonomy of Viruses (ICTV), renamed “severe acute respiratory syndrome coronavirus 2” (SARS-CoV-2) [66]. As the etiologic agent of the coronavirus disease 2019 (COVID-19) pandemic, SARS-CoV-2 has caused over 618.5 million human cases with more than 6.5 million deaths globally [67]. SARS-CoV-2 is an enveloped virus with a positive-sense single-stranded RNA genome that encodes structural, nonstructural, and accessory proteins [68]. All along the COVID-19 pandemic, the real-time RT-PCR procedure has been adopted by the WHO as a standard method for confirmation of acute SARS-CoV-2 infections due to its sensitivity and specificity [69]. Primers and probes for real-time RT-PCR detection of SARS-CoV-2 were designed to target and detect the genes encoding RdRp, envelope (E), and nucleocapsid (N) proteins [3]. The schematic diagram of SARS-CoV-2 structure, genome organization, and target genes for detection are illustrated in Figure 7.

A cycle threshold value <40 is interpreted as a positive detection of SARS-CoV-2 RNA. Nasopharyngeal/oropharyngeal swabs have typically been used to confirm the clinical diagnosis. Notably, SARS-CoV-2 could also be detected in specimens from other sites such as bronchoalveolar lavage fluid, sputum, fiber bronchoscope brush biopsy, feces, and blood [64,70]. The real-time PCR method was applied to detect SARS-CoV-2 in more than 64,000 specimens collected from COVID-19 suspects or individuals in contact tracing programs in Jakarta and neighboring areas, Indonesia, within the first year of the COVID-19 pandemic [64]. In order to assess environmental contamination with SARS-CoV-2 in a hospital setting, swab samples were collected from hospital surfaces such as intensive care unit (ICU) floors, medical floors, heating, ventilation, and air conditioning (HVAC) and then used for diagnosis of SARS-CoV-2 using RT-qPCR [71]. Similarly, environmental samples from surfaces of university classrooms, libraries, computer rooms, gymnasiums, and common areas have also been employed for the detection of SARS-CoV-2 RNA to evaluate the potency of SARS-CoV-2 transmission through indirect contact mediated by SARS-CoV-2 contaminated objects and surfaces [72]. To monitor the occurrence of SARS-CoV-2 RNA in wastewater as an indication that the community members shedding SARS-CoV-2 RNA in their stool, influent, secondary, and tertiary treated effluent water samples were collected and used for SARS-CoV-2 RNA detection using RT-qPCR [73].

In relation to COVID-19, the real-time RT-PCR assay has been applied to analyze the viral load dynamics in sputum and nasopharyngeal swabs of patients. The nucleotide sequences targeted for amplification were the SARS-CoV-2 open reading frame 1ab (ORF1ab) and N protein gene fragments. The viral load in the sputum was found to be higher than that in the nasopharyngeal swab at the time of disease presentation [48]. In addition, the viral load in the sputum samples decreased more slowly than in the nasopharyngeal swab samples as the disease progressed, primarily in patients with another underlying disease, such as hypertension or diabetes. These data suggested the value of using sputum specimens for SARS-CoV-2 detection to reduce the spread of COVID-19 within the community [48].

### 4.1. Detection of SARS-CoV-2 Variants

Like other RNA viruses, SARS-CoV-2 continuously mutates, resulting in the emergence of SARS-CoV-2 variants, which may have different pathological effects [74]. The variants B.1.617.2 (Delta), B.1.466.2, B.1.470, B.1.1.7 (Alfa), B.1.351 (Beta), P.1 (Gamma), P.2 (Zeta), and B.1.1.529 (Omicron) are among the most notable SARS-CoV-2 variants due to their potential to enhance biological threats [75,76,77,78]. Four mutations (N501Y, 69-70del (69/70 deletion), K417N, and E484K) in the spike protein may be linked to the potential biological effects of some of these variants. The real-time PCR method has also been applied as a fast and low-cost assay to detect SARS-CoV-2 mutations, thus facilitating the early process of decision-making to prevent the spread of SARS-CoV-2. Three tests were developed to detect spike (S) gene mutations of SARS-CoV-2 (N501Y, 69-70del, K417N, and E484K). Specific primers were designed and validated using nucleotide sequencing. The assays were applied to clinical samples from COVID-19 patients. The strategy was shown to allow the detection of the E484K mutation and the P.2 variant [76]. Similarly, a one-step real-time RT-PCR was developed to detect two mutations of concern, N501Y and E484K, in the SARS-CoV-2 S protein, which had been linked to enhanced viral transmissibility and immune escape, respectively. A 153 bp amplicon of the SARS-CoV-2 S gene, flanking both mutations, was targeted. The real-time RT-PCR assay was able to accurately identify the nucleotide changes associated with the E484K and N501Y substitutions of the SARS-CoV-2 S protein. The basic principles of the technique can be applied to develop similar assays for the detection of emerging mutations of concern [79]. A real-time RT-PCR assay was also designed to detect SARS-CoV-2 variants of concern by analyzing single nucleotide polymorphisms in the spike protein [80]. This user-friendly, cheap test was considered to be applicable for the rapid identification of prevalent SARS-CoV-2 variants of concern, such as the delta variant. The data generated can be used to supplement the data obtained by genomic sequencing [80].

### 4.2. Diagnosis of SARS-CoV-2

The real-time RT-PCR method has been considered to be the gold standard for the confirmatory test of SARS-CoV-2 infection [81]. However, it is critical to note that the use of real-time RT-PCR presents considerable challenges, and the results must be interpreted with caution. One of the crucial issues with real-time RT-PCR testing is the possibility of bringing about false-negative, and false-positive results, as a number of factors may cause inconsistency in real-time RT-PCR assays [81]. Because SARS-CoV-2 evolves rapidly, false-negative results can be due to mutation in the viral genome changing target sequences of primers and probes. Therefore, it potentially hinders the detection of the virus in samples from COVID-19-positive individuals, as seen in S-gene target failure cases [82]. Amplification of different target genes could be implemented in this scenario to improve the validity of the results. Variations in terms of the quality of the kits used, the skill of the laboratory personnel, sample types, and specimen conditions may also affect the results. The use of different specimen types (stool and blood) besides respiratory specimens has been proposed to avoid inconsistent results [81].

Incorrect negative results of COVID-19 real-time PCR testing have been described. From a study using a large sample size, the rate of false negative results was found of approximately 9.3% [83]. Thus, it is highly necessary to evaluate the performance of SARS-CoV-2 real-time RT-PCR assays in order to ensure the accuracy of COVID-19 detection. It should be noted that false negative results may have high implications as they may lead to positive case clusters [83]. Due to false negative results, infected individuals (who are possibly asymptomatic) might not be isolated and can therefore infect others [84]. False-negative results can also be due to low concentration of SARS-CoV-2 virus in patients, alteration in viral shedding, suboptimal specimen collection, testing too early in the disease process, low analytical sensitivity, and wrong specimen type [83]. Therefore, proper sampling protocols, good laboratory practice standards, and the use of high-quality extraction and real-time RT-PCR kits are of paramount importance to minimize the possibility of inaccurate results [81]. To improve sensitivity, a single-tube-nested real-time RT-PCR, employing two sets of primer (external and internal), was developed by manipulating annealing temperatures to permit the processes of reverse transcription, external primer, amplification, and internal primer amplification to occur sequentially in one tube. This novel and highly sensitive assay offered advantages in detecting SARS-CoV-2 in samples of low viral load, such as pooled clinical specimens and saliva samples [85].

In general, the high sensitivity of PCR-based molecular assays also makes them prone to false positive results, mainly due to contamination, as even a single copy of contaminant nucleic acid can undergo PCR amplification to a detectable positive signal [6]. False positive results can be due to carry-over of a previous amplicon of the same target sequence, reagent contamination, sample cross-contamination, mislabeling of samples, and cross-reactions with other viruses or genetic material. Cross-contamination from a positive clinical sample to a negative one can take place during specimen sampling, handling, processing, or analysis [6,86,87,88].

False positive results during COVID-19 testing using the real-time RT-PCR method have been reported with a rate of 0.5% [88]. It is important to note that a false positive result wrongly labels an individual as being infected with COVID-19 [84]. Any false positive COVID-19 results may have adverse impacts, such as overestimation of the COVID-19 incidence, unnecessary treatment, and investigation, wasting time and resources for unneeded isolation and contact tracing, the individual being placed with other inpatients with COVID-19 and consequently exposed to SARS-CoV-2, delayed surgery and prolongment of hospital stays [88]. Unfortunately, in most settings, grouping patients with positive results is unavoidable during periods of very high viral prevalence [88]. Other potential impacts of false positive results include distress, enforced isolation and stigmatization, fear of infecting others, travel cancelation, and loss of income [89]. In order to minimize the risk of false positive results during COVID-19 testing, it is important to increase awareness of false positives, have skilled and well-trained personnel, and improve laboratory procedures for sample collection and testing. In addition, the diagnostic results must be carefully interpreted at all times [88].

### 4.3. Viral Load Analysis

For the determination of viral load, it is critical to closely observe variations among different runs. Notably, the Ct value itself cannot be directly interpreted as viral load without a standard curve using reference materials. A good standard curve with an acceptable limit of detection is needed for accurate viral load analysis. The validity of the standard curve using reference materials, or plasmid controls with known viral copy numbers, should be confirmed in order to interpret Ct values in terms of viral loads [90]. Related to the use of Ct value to declare whether a person is COVID-19 positive or negative, it is critical to note that many factors may influence the real-time RT-PCR, hence, the resultant Ct value. These factors include sample type, stability of RNA molecules during sampling, storage, and RNA extraction, the efficiency of the RNA isolation process, the presence of inhibiting compounds, and the efficiency of reverse transcription [91]. Therefore, the Ct value is a relative value, not an absolute one, and for this reason, it must be interpreted with caution. It was proposed that digital PCR may play a role as a confirmatory tool to augment the interpretation of real-time RT-PCR Ct values in SARS-CoV-2 diagnosis [91]. It was reported that the Ct value is not correlated with disease severity [92,93]. Furthermore, droplet digital PCR (ddPCR), which enables accurate quantification of SARS-CoV-2 viral load from crude lysate without nucleic acid purification, has been developed [94]. This technique may provide absolute viral counts without the need for a standard curve, hence, simplifying the COVID-19 testing [94]. Similarly, a one-step multiplex droplet digital RT-PCR assay has also been developed for sensitive quantification of SARS-CoV-2 RNA. This novel method permits the simultaneous detection of SARS-CoV-2 E, RdRp, and N genes [95].

The real-time PCR method has also been applied to detect COVID-19 in the environment. When an infected individual sneezes or coughs, the respiratory droplets or aerosol settle down on the environment’s surfaces [96]. Contamination may also take place when an infected individual comes in direct contact with such surfaces. For environment COVID-19 testing, samples can be taken from isolation rooms, healthcare settings, and quarantine rooms. Detection of SARS-CoV-2 in the environment is critical to obtain data on the persistence of the virus in the air or on surfaces, the extent of contamination, and how air and surfaces become contaminated [96]. Furthermore, the real-time RT-PCR technique has also been successfully used to detect SARS-CoV-2 infection in canines (dogs), confirming instances of human-to-animal transmission of SARS-CoV-2 [97].

Apart from real-time RT-PCR, lateral flow immunoassays are rapid, low-cost, portable, and easy-to-use assays for COVID-19 testing and have been developed and evaluated all over the world [98,99]. In principle, these assays work by the binding of conjugated antibodies to a specific antigen in a sample. The main target antigens are the immunogenic proteins of SARS-CoV-2, such as the S (spike) protein, which is the most exposed, and the N protein, which is abundantly expressed during infection [100]. The lateral flow immunoassays will be helpful in accelerating COVID-19 screening if they show the same sensitivity and specificity as real-time RT-PCR tests [98]. A systematic review assessing the sensitivity and specificity of 24 papers reporting the use of lateral flow immunoassays in the detection of SARS-CoV-2, which in total involved more than 26,000 test results, indicated that the performance of the lateral flow immunoassays developed for COVID-19 testing was heterogeneous depending on the kit manufacturer with sensitivity ranging from 37.7% to 99.2% and specificity ranging from 92.4% to 100.0% [101]. Notably, several studies have demonstrated that the lateral flow immunoassays for SARS-CoV-2 antigen detection show comparable sensitivity and specificity with the real-time RT-PCR assay and these researchers, therefore, concluded that these rapid and simple tests have the potential to be applied as screening assays, particularly in a high prevalence area of infection [98,99,102].

## 5. Conclusions

Real-time PCR is a modification of the conventional PCR technique, enabling real-time monitoring of the PCR progress. The real-time PCR systems are dependent on a fluorescent reporter of the probe used for detection and quantification. It is a powerful technique that offers great sensitivity and specificity and can be used for both qualitative and quantitative analysis. It has revolutionized molecular methods and become a common tool for detecting and quantifying expression profiles of numerous selected genes. The real-time PCR technique has been widely applied in different research areas for various types of analysis of biological samples. In the context of the COVID-19 pandemic, the real-time RT-PCR assay has been considered the gold standard for confirmation of SARS-CoV-2 infection. Future studies should focus on developing low-cost, portable, and user-friendly instruments suitable for application in remote and resource-limiting settings. Improved quality of reagents and standardized protocols are critical to avoid invalid negative and false positive results. Further development of the multiplexing strategy is also critical to allow the effective identification of multiple genes.

## Figures and Tables

**Figure 1 genes-13-02387-f001:**
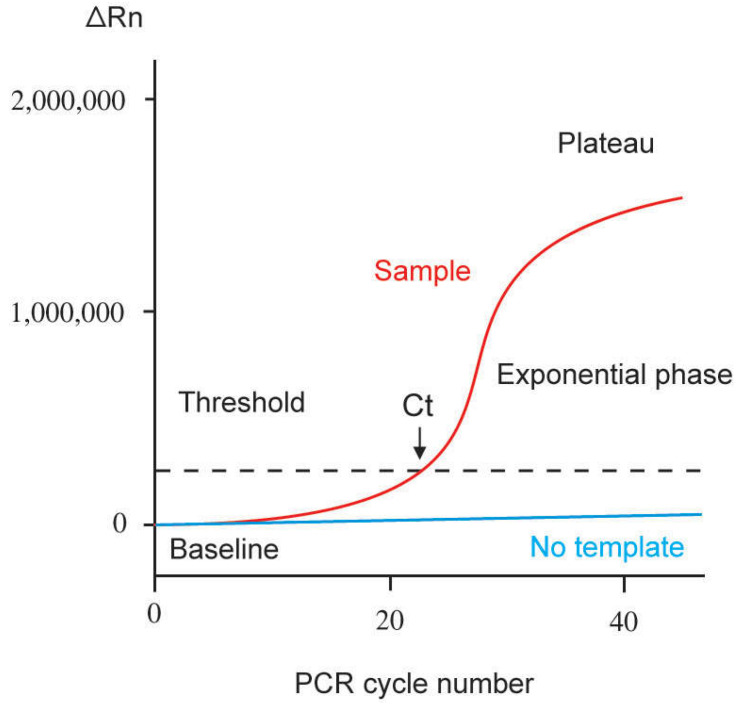
Representation of a single amplification plot of real-time quantitative PCR. ∆Rn = fluorescence emission of the amplicon at each time point minus fluorescence emission of the baseline. Ct = threshold cycle. Baseline refers to the PCR cycles in which the fluorescent signal of a reporter accumulates. However, it is below the limits of detection of the instrument (adapted from Arya et al. [5]).

**Figure 2 genes-13-02387-f002:**
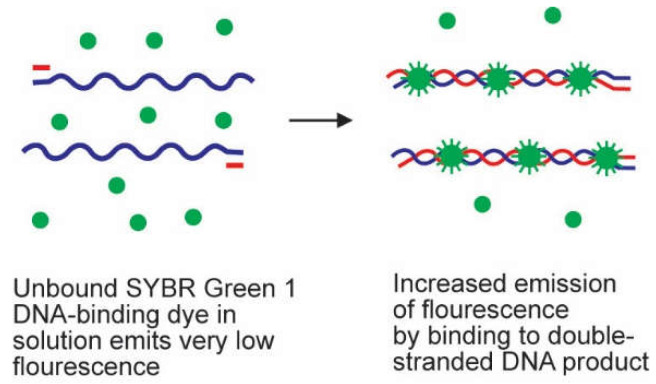
Mechanism of action of SYBR Green 1 dye. SYBR Green 1 probe is a double-stranded DNA-intercalating agent which exhibits very little fluorescence whilst free in solution. In the time of primer elongation and polymerization, SYBR Green 1 molecules become inserted into the double-stranded amplicons, causing an increase in detectable fluorescence [5].

**Figure 3 genes-13-02387-f003:**
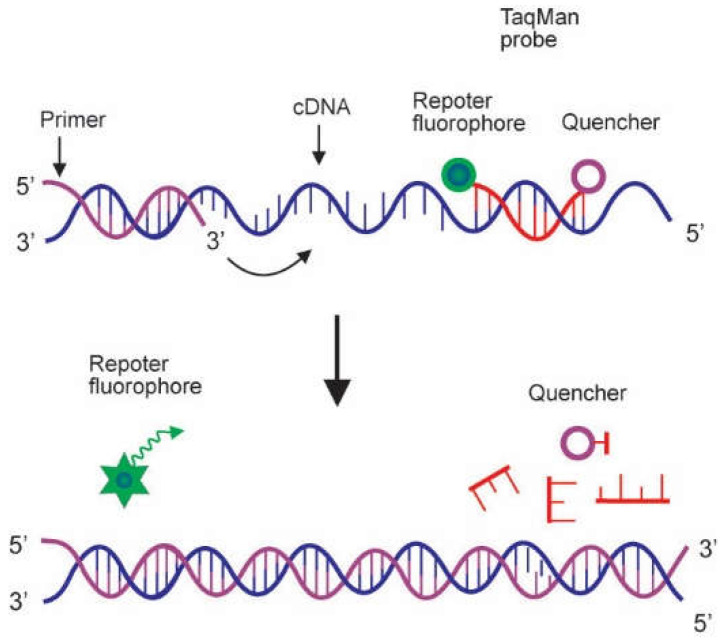
Mode of action of TaqMan probe. The TaqMan probe is a hydrolysis probe with a fluorescent reporter dye bound to its 5′ end and a quencher dye at its 3′ terminus. Whilst the probe is intact, fluorescence resonance energy transfer (FRET) occurs, and the fluorescence emission of the reporter dye is absorbed by the quenching dye. In the presence of the target sequence, the fluorogenic probe anneals downstream from one of the primer sites. It is cleaved by the 5′ nuclease activity of the *Taq* polymerase enzyme during the elongation step of the real-time PCR. Cleavage of the probe by *Taq* polymerase during PCR segregates the reporter and quencher dyes, thereby producing a fluorescence signal (Adapted from Arya et al. [5]).

**Figure 4 genes-13-02387-f004:**
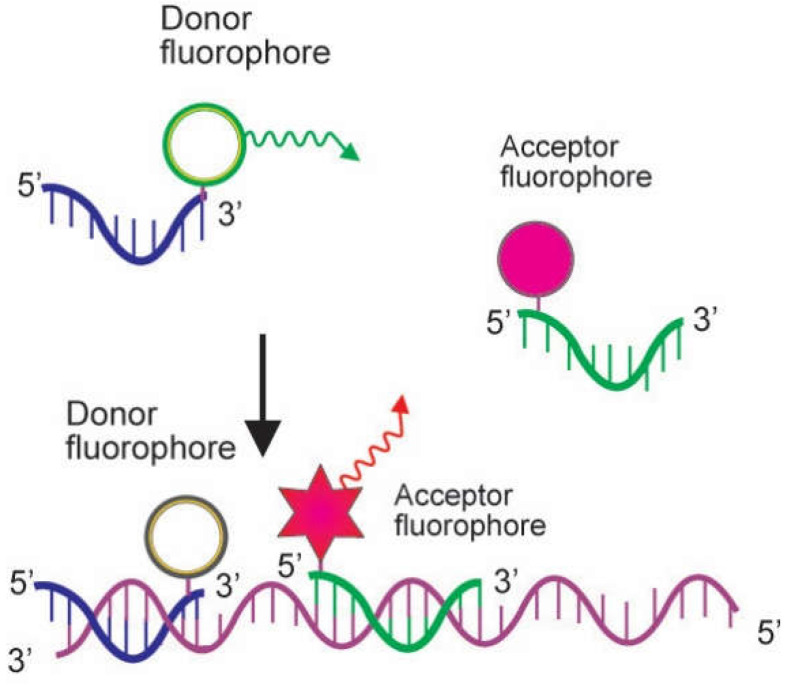
Action mode of dual hybridization probe. The dual hybridization probe consists of two hybridization probes, one brings a donor fluorophore at its 3′ end, and the other is labeled with an acceptor fluorophore at its 5′ terminus. After the denaturation phase, both probes attach to their target sequence in a head-to-tail arrangement during the annealing step. This causes the two dyes in close proximity to facilitate fluorescence resonance energy transfer (FRET). The donor dye in one of the probes transmits energy, facilitating the other one to dissipate fluorescence at a distinct wavelength (Adapted from Arya et al. [5]).

**Figure 5 genes-13-02387-f005:**
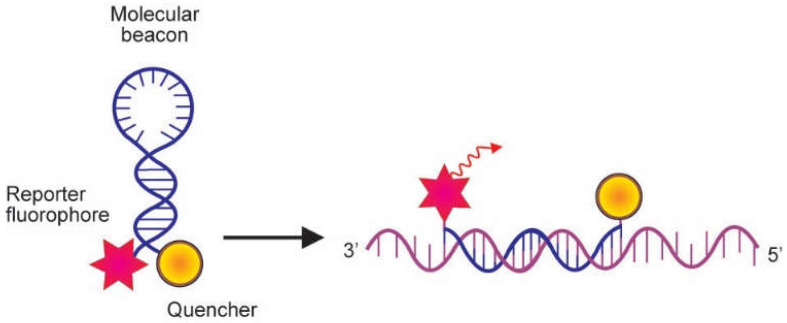
Mechanism of action of the molecular beacon. Molecular beacons contain covalently linked fluorescent and quenching dyes at either end of a single-stranded DNA molecule. Whilst free in solution, the probe is maintained in a hairpin conformation by complementary stem sequences at both ends of the probe, which brings the fluorescent dye and the quencher in close proximity. This causes fluorescence resonance energy transfer (FRET) to occur, which suppresses reporter fluorescence. The loop part of the molecule is complementary to the target nucleic acid molecule. In the presence of a target sequence, the loop hybridizes to the complementary target sequence throughout the annealing step, resulting in a conformational alteration that forces the reporter and quencher dyes to separate, and fluorescence is emitted (Adapted from Arya et al. [5]).

**Figure 6 genes-13-02387-f006:**
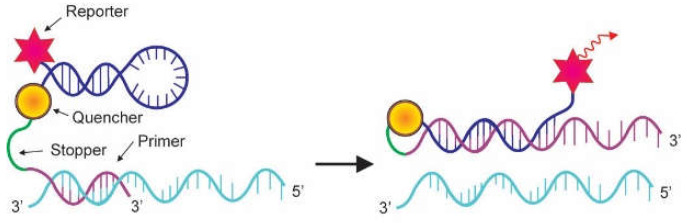
Mechanism of action of Scorpion probe. The scorpion probe adopts a stem-and-loop conformation held by complementary stem sequences on the 5′ and 3′ sides of the probe. A fluorophore is attached to the 5′ end, and a quencher is linked to the 3′ end of the probe. A specific probe sequence is held within the hairpin loop, which is linked to the 5′ terminus of a PCR primer sequence by a PCR stopper. This chemical variation hinders PCR from amplifying the stem-loop sequence of the scorpion primer. In the course of PCR, scorpion primer is elongated to generate an amplicon. In the annealing phase, the specific probe sequence in the scorpion tail curls back to hybridize with the complementary target sequence in the amplicon, hence opening up the hairpin loop. This prevents the fluorescence from being quenched, and a signal is detected (Adapted from Arya et al. [5]).

**Figure 7 genes-13-02387-f007:**
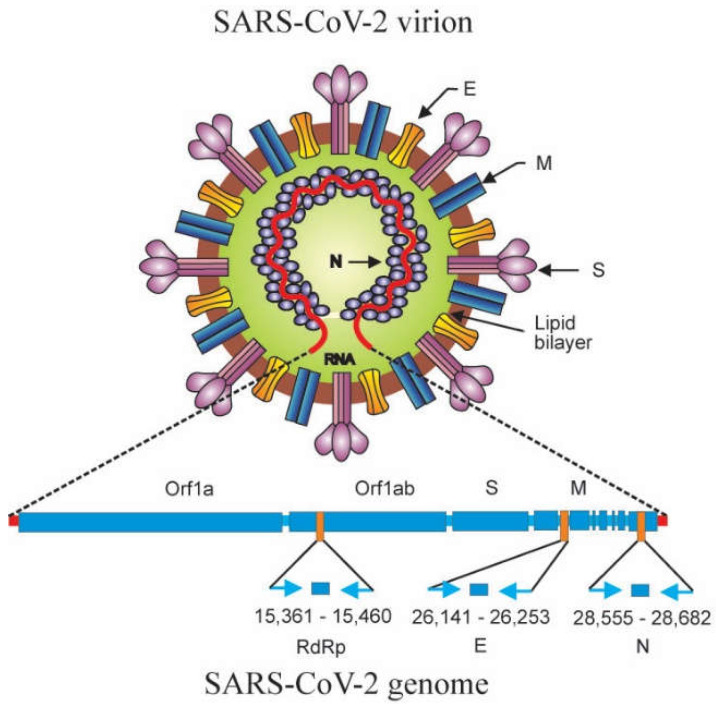
Schematic diagram of molecular structure, genome organization, and relative positions of amplicon targets on the SARS-CoV-2 genome. On the SARS-CoV-2 virion, together with membrane protein and envelope protein, the spike protein glycoprotein projects from a lipid bilayer, giving the virion a distinctive appearance. SARS-CoV-2 virion (top): M: membrane protein; E: envelope protein; S: spike protein; N: nucleocapsid protein. The SARS-CoV-2 genomic RNA is associated with the nucleocapsid protein forming the ribonucleoprotein with a helical structure. The SARS-CoV-2 genome encodes structural (S, M, E, N) and nonstructural proteins. The relative positions of amplicon targets (RdRp, E, N) are shown. SARS-CoV-2 genome (bottom): Orf1a: open reading frame 1a; Orf1ab: open reading frame 1ab; S: spike protein gene; M: membrane protein gene; RdRp: RNA-dependent RNA polymerase gene; E: envelope protein gene; N: nucleocapsid protein gene (adapted from Corman et al. [3]; Artika et al. [55]).

**Table 1 genes-13-02387-t001:** Examples of application of real-time polymerase chain reaction.

Field	Application	References
Gene expression analysis	Analysis of wireless fidelity radiofrequency radiation on the expression of *E. coli* genes that potentially alter its pathogenic traits	[23]
	Examination of plant gene expression impacting lignin synthesis for plant cell wall structure	[24]
	Analysis of gene expression as a potential biomarker for early-stage diagnosis in colorectal tumor and cancer patients	[25]
	Examination of microRNA expression profile in response to viral infection	[26]
Detection of mutation	Detection of mutation patterns in human cancer cells	[27,28]
	Detection and quantitative analysis of mutation for monitoring drug resistance	[29,30]
Food Analysis	Detection of genetically modified organisms (GMO)	[31,32,33]
	Detection of allergens in food	[34]
	Detection of pork in food products	[35]
Bioremediation monitoring	Monitoring microbial degradation	[36,37,38,39]
Detection and quantification of pathogens	Detection of pathogenic bacteria	[40]
	Identification of microbial species as etiology of a disease	[41]
	Molecular bacterial load assay (i.e., *Mycobacterium tuberculosis*)	[42]
	Determination of growth fitness of plasmodium parasites	[43]
	Detection of pathogenic RNA viruses	[3,44,45]
	Diagnosis of pathogenic DNA viruses	[46]
	Analysis of viral load associated with clinical features of the disease	[8,47,48]

**Table 2 genes-13-02387-t002:** Examples of some reference genes commonly applied for analysis of gene expression.

Gene Name	Abbreviation	Application	Reference
Glyceraldehide-3-phosphate dehydrogenase	*GADPH*	Analysis of gene expression in human cell lines, human airway epithelial cells, wound healing model, human skeletal muscle tissue, human breast cells, induced pluripotent stem cell reprogramming	[49,52,53,54,55]
Actin, beta	*ACTB*	Analysis of gene expression in wound healing model, human skeletal muscle tissue, human breast cells	[53,54]
Ribosomal RNA 18S	*18S*	Analysis of gene expression in	[53,54]
		wound healing model, human skeletal muscle tissue, human breast cells	
β-2-microglobulin	*β-2M*	Analysis of gene expression in wound healing model, human skeletal muscle tissue, human breast cells	[53,54]
Phosphoglycerate kinase 1	*PGK1*	Analysis of gene expression in induced pluripotent stem cell reprogramming	[55]
Polyubiquitin C	*UBC*	Determination of gene expression in human cell lines	[49]
DNA topoisomerase 1	TOP1	Study of gene expression in human cell lines	[49]
ATP synthase subunit beta, mitochondrial	ATP5B	Elucidation of gene expression in human cell lines, induced pluripotent stem cell reprogramming	[49,55]
Cyclophilin A	PPIA	Examination of gene expression in human airway epithelial cells	[52]

## Data Availability

Not applicable.

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
