# Peer review of "Real-Time Polymerase Chain Reaction: Current Techniques, Applications, and Role in COVID-19 Diagnosis"

_genes, 2022, doi:10.3390/genes13122387_

Round 1

Reviewer 1 Report

Dear Editor,

Detection of emerging viruses has become increasingly important as a result of the COVID-19 pandemic. There is no doubt that real-time PCR is one of the cheapest and fastest methods of detecting SARS-CoV-2 mutations. In the current study, using the COVID-19 outbreak in Indonesia after 2019 as a starting point, the usage areas of real-time PCR have been reviewed in a very comprehensive manner. Despite the fact that the second part of this review and the aftermath of it are quite impressive, the abstract and introduction did not live up to common expectations. For example, there is no statement in the introduction of this study that clearly distinguishes it from other review articles. Additionally, although the researchers intended to appeal to researchers throughout the world in such a comprehensive review study, they chose to limit themselves to Indonesia in the introduction section of their work. Is it advisable to limit our studies in such a manner? Abstract, which appears to have been written separately from the study, has very general statements. The originality of the study should be emphasized more strongly in the abstract section. These parts could have been handled better by the authors. Since the deficiencies mentioned above may be corrected, it would be beneficial to publish it due to its contribution to the field.

Major comments

In accordance with Instructions for Authors, references should be numbered in chronological order, with a number or numbers in square brackets, such as [1] or [2,3], or [4–6]. It is therefore necessary to correct all references cited in the text.

In the text, all gene names and gene regions should be italicized.

Minor comments

L14: Please use “Abstract” instead of “ABSTRACT”.

L30: Please use “Introduction” instead of “INTRODUCTION”.

L52: Please use “Basic Principles” instead of “BASIC PRINCIPLES”.

L80-81: It is recommended that several references be cited.

L90, L108, L397: “Figure ” instead of “Fig.” Or “Fig”.

L167: Please use “Applications” instead of “APPLICATIONS”.

L181: “RT-qPCR” instead of “qPCR”. In conformity with L193, this abbreviation is used.

L204, L223: “Real-time PCR” or “Real-time RT-PCR”. The item should be checked.

L235: “mutation” instead of “Mutation”.

L244: Please use italics when writing “PIK3CA”.

L252-254: References should be provided.

L266-269: There is a need to verify the cited reference. This article does not make any statement regarding the markers mentioned in the cited source. It is possible that the author has written another article pertaining to these markers.

L275-278: References should be provided.

L283: Please use italics when writing  “cat, aadD or tet-l”

L284-286: References should be provided.

L285: Please check the spelling of "mat k, rpl16, and trnH-psbA". Italics should be used for these markers.

L286: Please check the spelling of "mat k”.

L300: Please use italics when writing “alkB4”.

L309-311: References should be provided.

L320-324: Reference should be added at the end of the sentence.

L324-326: References should be provided.

L337-338: References should be provided.

L379: Please use “Detction and Quantification ofSARS-CoV-2” instead of “DETECTION AND QUANTIFICATION OF SARS-CoV-2”.

L390-392: Repeated words.  “during COVID-19” should be removed.

L395: “[31]”. This format should be used for all references in the text.

L410-417: Please use “:” instead of “=”.

L428: Please use “variants” instead of “Variants”.

L442-446: References should be provided.

L450-451: References should be provided.

L457-458: References should be provided.

L460-462: References should be provided.

L471-472: The authors have indicated the source of the information in this type of sentence that states certainty.  It should be noted that most of the sentences I have mentioned as a source are similar to this one.

L530-532: References should be provided.

L536: Please use “E” instead of “envelope (E)”. It was previously abbreviated in L395.

L538-540: References should be provided.

L568: Please use “Conclusion” instead of “CONCLUSION”.

L569: Please use “PCR” instead of “polymerase chain reaction”. It was previously abbreviated in L31.

L584-588: It is appropriate to abbreviate author names.

L594: Please use “References” instead of “REFERENCES:”.

Author Response

Thank you for processing our manuscript (genes-2025187) and for forwarding the reviewer’s comments.

We have now completed the revision of the manuscript.  The manuscript has been revised in accordance with the reviewer’s comments as follows:

Response to Reviewer 1:

L14: Please use Abstract instead of ABSTRACT.

Authors response: done

L30: Please use Introduction instead of INTRODUCTION.

Authors response: done

L52: Please use Basic Principles instead of BASIC PRINCIPLES.

Authors response: done

L80-81: It is recommended that several references be cited.

Authors response: a reference has been added

L90, L108, L397: Figure instead of Fig. Or Fig.

Authors response: done

L167: Please use Applications instead of APPLICATIONS.

Authors response: done

L181: RT-qPCR instead of qPCR. In conformity with L193, this abbreviation is used.

Response: done

L204, L223: Real-time PCR or Real-time RT-PCR. The item should be checked.

Response: “Real-time PCR”, because referring to the process after cDNA construction

L235: mutation instead of Mutation.

Response: done

L244: Please use italics when writing PIK3CA.

Response: done

L252-254: References should be provided.

Response: done

L266-269: There is a need to verify the cited reference. This article does not make any statement regarding the markers mentioned in the cited source. It is possible that the author has written another article pertaining to these markers.

Authors’ response:  Thank you for the correction. The accurate reference (Fraiture et al., 2015 has been cited, replacing Fraiture et al., 2021).

L275-278: References should be provided.

Response: done

L283: Please use italics when writing cat, aadD or tet-l

Response: done

L284-286: References should be provided.

Response: done

L285: Please check the spelling of "mat k, rpl16, and trnH-psbA". Italics should be used for these markers.

Response: has been checked.  In the original paper, “not italics” for markers.

L286: Please check the spelling of "mat k”.

Response: has been checked

L300: Please use italics when writing “alkB4”.

Response: done

L309-311: References should be provided.

Response: done

L320-324: Reference should be added at the end of the sentence.

Response: done

L324-326: References should be provided.

Response: done

L337-338: References should be provided.

Response: done

L379: Please use “Detection and Quantification of SARS-CoV-2” instead of “DETECTION AND QUANTIFICATION OF SARS-CoV-2”.

Response: done

L390-392: Repeated words.  “during COVID-19” should be removed.

Response: done

L395: “[31]”. This format should be used for all references in the text.

Response: done

L410-417: Please use : instead of =.

Response: done

L428: Please use “variants” instead of “Variants”.

Authors response: done

L442-446: References should be provided.

Response: done

L450-451: References should be provided.

Response: done

L457-458: References should be provided.

Authors response: done

L460-462: References should be provided.

Authors response: done

L471-472: The authors have indicated the source of the information in this type of sentence that states certainty.  It should be noted that most of the sentences I have mentioned as a source are similar to this one.

Authors response: Noted.

L530-532: References should be provided.

Authors response: done

L536: Please use “E” instead of “envelope (E)”. It was previously abbreviated in L395.

Authors response: done

L538-540: References should be provided.

Authors response: done

L568: Please use “Conclusion” instead of “CONCLUSION”.

Authors response: done

L569: Please use “PCR” instead of “polymerase chain reaction”. It was previously abbreviated in L31.

Authors response: done

L584-588: It is appropriate to abbreviate author names.

Authors response: done

 L594: Please use “References” instead of “REFERENCES:”.

Authors response: done

Reviewer 2 Report

Real-time PCR has become one of the most powerful molecular techniques and is widely used in biological sciences and medicine because it is quantitative, accurate, sensitive, and rapid. Current applications of real-time PCR included in manuscript are analysis of gene expression, detection of mutation, detection and quantification of pathogen, detection of genetically modified organism, detection of allergen, monitoring of microbial degradation, species identification and determination of parasite fitness.

The overall strengths of the article are pretty obvious. First, it helps readers with even basic knowledge to understand the principles and applications of PCR methodology. Second, it illustrates many facts and key points in the technique, providing readers with possible solutions. Third, the information providing to the readers about innovative and/or contributive methods in the context is extremely helpful for work that is more precise.

Abstract clearly illustrated the aims of this article. However, if it can provide more specific words in the title or/and the abstract, it would prompt readers to take a few minutes more to reconsider whether or not he/she should spend time reading the whole article. The proposal is related to the fact that the authors are described in much more details the applications and examples for SARS-CoV-2 detection and variants identification.

The study covers a good number of references, most of the last few years but I think there is a sufficient amount of information that would help the authors to expand their horizons and supplement the data in the article. (Example: in chapter 3.2, 3.4 and 4 have only three-four works are cited within the whole chapter). Those that are considered by the authors, on what principle are they selected?

Here are some other minor comments, that authors need to go through them:

1.      Chapter 2 - This chapter describes the basic principles of real-time polymerase chain reaction. To highlight and illustrate the methodology, a general scheme/figure of the methodology can be given (not only of the model of a single amplification plot of real time quantitative PCR) and even the modifications of the PCR technique.

2.      Chapter  3 - The text includes table 1 with examples from recent publications relate to plants, animal models, cell cultures, and viruses. On what principle/criteria are they selected to be included or not in table?

3.      Chapter 3.1. – Other genes (house-keeping) used as endogenous controls may be mentioned  in context – GADPH, gene encoding ribosome 18S rRNA subunit, NADPH, Actin2…). You can even make a table with the most frequently used ones depending on the research objects or conditions.

4.      Names of the genes need to be written in italics (line 195,283, 300, 340,  ….). Correct in the text if there are such technical errors somewhere.

5.      References are missing – line 351, 366, 422, 523, 532….

6.      Chapter 4 - Line 400 -  …”Notably, SARS-CoV-2 could also be detected in specimens  from other sites such as bronchoalveolar lavage fluid, sputum, fibrobronchoscope brush biopsy, feces, and blood”…………. You may also mention its discovery in wastewater in worldwide.

7.      Chapter 4.1.  – line 431 - if we follow the context it is good to note the name of the other SARS-CoV-2 variants – B.1.1.7 (Alfa), B.1.351(Beta), P.1 (Gamma), P.2 (Zeta)..

8.      Chapter 4.2 – Is there data how many percent of the test probes are false-positive and false-negative?

In general, the work is a significant contribution to the field and sounds scientifically. The manuscript presents very up-to-date and detailed information on the real-time PCR methodology. It is complex and supposes а very good knowledge not only of the literary sources, but also of the practical side of the methodology

It can be enriched if it is supported with additional figures/shames or/and tables. All these refinements will make the article more interesting for readers from multiple backgrounds. It was a great pleasure for me to read it and give many small directions for its improvement.

Author Response

Bogor, 1 December 2022

Dear Editor-in-Chief

Genes

Thank you for processing our manuscript (genes-2025187) and for forwarding the reviewer’s comments.

We have now completed the revision of the manuscript.  The manuscript has been revised in accordance with the reviewer’s comments as follows:

Bogor, 1 December 2022

Dear Editor-in-Chief

Genes

Thank you for processing our manuscript (genes-2025187) and for forwarding the reviewer’s comments.

We have now completed the revision of the manuscript.  The manuscript has been revised in accordance with the reviewer’s comments as follows:

Comments from Reviewer 2:

Real-time PCR has become one of the most powerful molecular techniques and is widely used in biological sciences and medicine because it is quantitative, accurate, sensitive, and rapid. Current applications of real-time PCR included in manuscript are analysis of gene expression, detection of mutation, detection and quantification of pathogen, detection of genetically modified organism, detection of allergen, monitoring of microbial degradation, species identification and determination of parasite fitness.

The overall strengths of the article are pretty obvious. First, it helps readers with even basic knowledge to understand the principles and applications of PCR methodology. Second, it illustrates many facts and key points in the technique, providing readers with possible solutions. Third, the information providing to the readers about innovative and/or contributive methods in the context is extremely helpful for work that is more precise.

Authors response: Thank you for the positive comments and appreciation

Abstract clearly illustrated the aims of this article. However, if it can provide more specific words in the title or/and the abstract, it would prompt readers to take a few minutes more to reconsider whether or not he/she should spend time reading the whole article. The proposal is related to the fact that the authors are described in much more details the applications and examples for SARS-CoV-2 detection and variants identification.

Authors response: Title and abstract have been revised to indicate discussion related to SARS-CoV-2 detection.

The study covers a good number of references, most of the last few years but I think there is a sufficient amount of information that would help the authors to expand their horizons and supplement the data in the article. (Example: in chapter 3.2, 3.4 and 4 have only three-four works are cited within the whole chapter). Those that are considered by the authors, on what principle are they selected?

Authors response: they are selected based on the most recent and most related to the topics of discussion, for examples references on detection of mutation using real-time PCR are selected (those using other methods are not considered).  Discussion has been extended and some more references have been cited.

Here are some other minor comments, that authors need to go through them:

  1. Chapter 2 - This chapter describes the basic principles of real-time polymerase chain reaction. To highlight and illustrate the methodology, a general scheme/figure of the methodology can be given (not only of the model of a single amplification plot of real time quantitative PCR) and even the modifications of the PCR technique.

Authors response: discussion has been extended and some figures have been added.

  1. Chapter 3 - The text includes table 1 with examples from recent publications relate to plants, animal models, cell cultures, and viruses. On what principle/criteria are they selected to be included or not in table?

Authors response: They are selected based on the most recent and related publication to reflect the wide ranges of real-time PCR applications with priority on studies conducted in Indonesia or involving the authors.

  1. Chapter 3.1. Other genes (house-keeping) used as endogenous controls may be mentioned in context GADPH, gene encoding ribosome 18S rRNA subunit, NADPH, Actin2). You can even make a table with the most frequently used ones depending on the research objects or conditions.

Authors response: a table on the most commonly used reference genes has been added.

  1. Names of the genes need to be written in italics (line 195,283, 300, 340, .). Correct in the text if there are such technical errors somewhere.

Authors response: done

  1. References are missing line 351, 366, 422, 523, 532.

Authors response: References have been provided

  1. Chapter 4 - Line 400 - …”Notably, SARS-CoV-2 could also be detected in specimens  from other sites such as bronchoalveolar lavage fluid, sputum, fibrobronchoscope brush biopsy, feces, and blood”…………. You may also mention its discovery in wastewater in worldwide.

Authors response: discussion has been extended, more references have been cited.

  1. Chapter 4.1.  line 431 - if we follow the context it is good to note the name of the other SARS-CoV-2 variants B.1.1.7 (Alfa), B.1.351(Beta), P.1 (Gamma), P.2 (Zeta)..

Authors response: done

  1. Chapter 4.2 Is there data how many percent of the test probes are false-positive and false-negative?

Authors response: have been mentioned (data added).

In general, the work is a significant contribution to the field and sounds scientifically. The manuscript presents very up-to-date and detailed information on the real-time PCR methodology. It is complex and supposes а very good knowledge not only of the literary sources, but also of the practical side of the methodology

Authors response: Thank you.

It can be enriched if it is supported with additional figures/shames or/and tables. All these refinements will make the article more interesting for readers from multiple backgrounds. It was a great pleasure for me to read it and give many small directions for its improvement.

Authors response: Done.  Some figures have been added and a table has been added.

Thank you for considering our article and for the good cooperation.

With best regards,

Prof. I Made Artika

Department of Biochemistry,

Bogor Agricultural University.

Round 2

Reviewer 1 Report

Dear Authors,

I have been carefully reviewed your revised article with the id number "genes-2025187". In my opinion, this revised article incorporates all of the points raised in the original draft to the best of my knowledge. This review will contribute significantly to the theoretical literature regarding real-time PCR's use in detecting emerging viruses, including SARS-CoV-2. Best wishes to all of the authors who contributed to the production of this wonderful work and congratulations on their future endeavors.
